# Improving GFlowNets for Text-to-Image Diffusion Alignment

**Dinghuai Zhang**,* **Yizhe Zhang, Jiatao Gu, Ruixiang Zhang, Josh Susskind,**
**Navdeep Jaitly, Shuangfei Zhai**
*Apple*

**Reviewed on OpenReview:** *https://openreview.net/forum?id=XDbY3qhM42*

## Abstract

Diffusion models have become the *de-facto* approach for generating visual data, which are trained to match the distribution of the training dataset. In addition, we also want to control generation to fulfill desired properties such as alignment to a text description, which can be specified with a black-box reward function. Prior works fine-tune pretrained diffusion models to achieve this goal through reinforcement learning-based algorithms. Nonetheless, they suffer from issues including slow credit assignment as well as low quality in their generated samples. In this work, we explore techniques that do not directly maximize the reward but rather generate high-reward images with relatively high probability — a natural scenario for the framework of generative flow networks (GFlowNets). To this end, we propose the **D**iffusion **A**lignment with **G**FlowNet (DAG) algorithm to post-train diffusion models with black-box property functions. Extensive experiments on Stable Diffusion and various reward specifications corroborate that our method could effectively align large-scale text-to-image diffusion models with given reward information. Our code is public available at https://github.com/apple/ml-diffusion-alignment-gflownet.

## 1 Introduction

Diffusion models (Sohl-Dickstein et al., 2015; Ho et al., 2020) have drawn significant attention in machine learning due to their impressive capability to generate high-quality visual data and applicability across a diverse range of domains, including text-to-image synthesis (Rombach et al., 2021), 3D generation (Poole et al., 2022), material design (Yang et al., 2023), protein conformation modeling (Abramson et al., 2024), and continuous control (Janner et al., 2022). These models, through a process of gradually denoising a random distribution, learn to replicate complex data distributions, showcasing their robustness and flexibility. The traditional training of diffu-

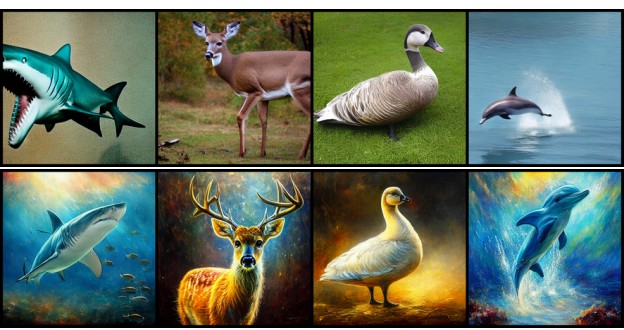

Figure 1: Generated samples before (top) and after (bottom) the proposed training with Aesthetic reward.

sion models typically relies on large datasets, from which the models learn to generate new samples that mimic and interpolate the observed examples.

However, such a dataset-dependent approach often overlooks the opportunity to control and direct the generation process towards outputs that not only resemble the training data but also possess specific, desirable properties (Lee et al., 2023). These properties are often defined through explicit reward functions that assess certain properties, such as the aesthetic quality of images. Such a requirement is crucial in fields where adherence to particular characteristics is necessary, such as alignment or drug discovery. The need to integrate explicit guidance without relying solely on datasets presents a unique challenge for training methodologies.

---

*Work done during internship at Apple MLR. Correspondence to `dinghuai233@gmail.com`

Previous works have utilized methods such as reinforcement learning (RL) (Black et al., 2023; Fan et al., 2023) to tackle this problem. Nonetheless, these methods still suffer from issues like low sample efficiency.

In this work, we propose a novel approach, diffusion alignment with GFlowNets (DAG), that fine-tunes diffusion models to optimize black-box reward functions directly. Generative flow networks (Bengio et al., 2023, GFlowNets), initially introduced for efficient probabilistic inference with given densities in structured spaces, provide a unique framework for this task. Though initially proposed for composite graph-like structures, prior works have extended the GFlowNet framework to diffusion modeling (Zhang et al., 2022a; Lahlou et al., 2023). This work further investigates GFlowNet-inspired algorithms for the task of text-to-image diffusion alignment. By aligning the learning process to focus on generating samples with probability proportional to reward functions rather than maximizing them, our method allows the diffusion model to directly target and generate samples that are not only high in quality but also fulfill specific predefined criteria. Besides developing a denoising diffusion probabilistic model-specific GFlowNet algorithm, we also propose a new KL-based way to optimize our models. In summary, our contributions are as follows:

- We propose Diffusion Alignment with GFlowNet (DAG), a GFlowNet-based algorithm using the denoising structure of diffusion models, to improve large-scale text-to-image alignment with a black-box reward function.

- We propose a KL-based objective for optimizing GFlowNets that achieves comparable or better sample efficiency. We further called the resulting algorithm for the alignment problem DAG-KL.

- Our methods achieve better sample efficiency than the reinforcement learning baseline within the same number of trajectory rollouts, as well as a better reward-diversity trade-off, across a number of different learning targets.

## 2 Preliminaries

### 2.1 Diffusion models

Denoising diffusion model (Vincent, 2011; Sohl-Dickstein et al., 2015; Ho et al., 2020; Song et al., 2020) is a class of hierarchical latent variable models. The latent variables are initialized from a white noise $\mathbf{x}_T \sim \mathcal{N}(\mathbf{0}, \mathbf{I})$ and then go through a sequential denoising (reverse) process $p_{\boldsymbol{\theta}}(\mathbf{x}_{t-1}|\mathbf{x}_t)$. Therefore, the resulting generated distribution takes the form of

$$p_{\boldsymbol{\theta}}(\mathbf{x}_0) = \int p_{\boldsymbol{\theta}}(\mathbf{x}_{0:T}) \, \mathrm{d}\mathbf{x}_{1:T} = \int p(\mathbf{x}_T) \prod_{t=1}^{T} p_{\boldsymbol{\theta}}(\mathbf{x}_{t-1}|\mathbf{x}_t) \, \mathrm{d}\mathbf{x}_{1:T}. \tag{1}$$

On the other hand, the variational posterior $q(\mathbf{x}_{1:T}|\mathbf{x}_0)$, also called a diffusion or forward process, can be factorized as a Markov chain $\prod_{t=1}^{T} q(\mathbf{x}_t|\mathbf{x}_{t-1})$ composed by a series of conditional Gaussian distributions $q(\mathbf{x}_t|\mathbf{x}_{t-1}) = \mathcal{N}(\mathbf{x}_t; \alpha_t/\alpha_{t-1}\mathbf{x}_{t-1}, (1 - \alpha_t^2/\alpha_{t-1}^2)\mathbf{I})$, where $\{\alpha_t, \sigma_t\}_t$ is a set of pre-defined signal-noise schedule. Specifically, in Ho et al. (2020) we have $\alpha_t^2 + \sigma_t^2 = 1$. The benefit of such a noising process is that its marginal has a simple close form: $q(\mathbf{x}_t|\mathbf{x}_0) = \int q(\mathbf{x}_{1:t}|\mathbf{x}_0) \, \mathrm{d}\mathbf{x}_{1:t-1} = \mathcal{N}(\mathbf{x}_t; \alpha_t\mathbf{x}_0, \sigma_t^2\mathbf{I})$.

Given a data distribution $p_{\mathrm{data}}(\cdot)$, the variational lower bound of model log likelihood can be written in the following simple denoising objective:

$$\mathcal{L}_{\mathrm{denoising}}(\boldsymbol{\theta}) = \mathbb{E}_{t,\mathbf{x}_0 \sim p_{\mathrm{data}}, \boldsymbol{\epsilon} \sim \mathcal{N}(\mathbf{0},\mathbf{I})} \left[ \|\mathbf{x}_0 - \hat{\mathbf{x}}_{\boldsymbol{\theta}}(\alpha_t\mathbf{x}_0 + \sigma_t\boldsymbol{\epsilon}, t)\|^2 \right], \tag{2}$$

where $\hat{\mathbf{x}}_{\boldsymbol{\theta}}(\mathbf{x}_t, t)$ is a deep neural network to predict the original clean data $\mathbf{x}_0$ given the noisy input $\mathbf{x}_t = \alpha_t\mathbf{x}_0 + \sigma_t\boldsymbol{\epsilon}$, which can be used to parameterize the denoising process $p_{\boldsymbol{\theta}}(\mathbf{x}_{t-1}|\mathbf{x}_t) = \mathcal{N}(\mathbf{x}_{t-1}; (\sigma_{t-1}^2\alpha_t\mathbf{x}_t + (\alpha_{t-1}^2 - \alpha_t^2)\hat{\mathbf{x}}_{\boldsymbol{\theta}}(\mathbf{x}_t, t))/\sigma_t^2\alpha_{t-1}, (1 - \alpha_t^2/\alpha_{t-1}^2)\mathbf{I})$. In practice, the network can also be parameterized with noise prediction or v-prediction (Salimans & Ho, 2022). The network architecture usually has a U-Net (Ronneberger et al., 2015) structure.

In multimodal applications such as text-to-image tasks, the denoising diffusion model would have a conditioning $\mathbf{c}$ in the sense of $p_{\boldsymbol{\theta}}(\mathbf{x}_0; \mathbf{c}) = \int p(\mathbf{x}_T) \prod_{t=1}^{T} p_{\boldsymbol{\theta}}(\mathbf{x}_{t-1}|\mathbf{x}_t; \mathbf{c}) \, \mathrm{d}\mathbf{x}_{1:T}$. The data prediction network, $\hat{\mathbf{x}}_{\boldsymbol{\theta}}(\mathbf{x}_t, t, \mathbf{c})$ in this case, will also take $\mathbf{c}$ as a conditioning input. We ignore the notation of $\mathbf{c}$ without loss of generality.

## 2.2 GFlowNets

Generative flow network (Bengio et al., 2021, GFlowNet) is a high-level algorithmic framework of amortized inference, also known as training generative models with a given unnormalized target density function. Let $\mathcal{G} = (\mathcal{S}, \mathcal{A})$ be a directed acyclic graph, where $\mathcal{S}$ is the set of states and $\mathcal{A} \subseteq \mathcal{S} \times \mathcal{S}$ are the set of actions. We assume the environmental transition is deterministic, *i.e.*, one action would only lead to one next state. There is a unique *initial state* $\mathbf{s}_0 \in \mathcal{S}$ which has no incoming edges and a set of *terminal states* $\mathbf{s}_N$ without outgoing edges. A GFlowNet has a stochastic *forward policy* $P_F(\mathbf{s}'|\mathbf{s})$ for transition $(\mathbf{s} \to \mathbf{s}')$ as a conditional distribution over the children of a given state $\mathbf{s}$, which can be used to induce a distribution over trajectories via $P(\tau) = \prod_{n=0}^{N-1} P_F(\mathbf{s}_{n+1}|\mathbf{s}_n)$, where $\tau = (\mathbf{s}_0, \mathbf{s}_1, \ldots, \mathbf{s}_N)$. On the other hand, the *backward policy* $P_B(\mathbf{s}|\mathbf{s}')$ is a distribution over the parents of a given state $\mathbf{s}'$. The *terminating distribution* defined by $P_T(\mathbf{x}) = \sum_{\tau \to \mathbf{x}} P_F(\tau)$ is the ultimate terminal state distribution generated by the GFlowNet. The goal of training GFlowNet is to obtain a forward policy such that $P_T(\cdot) \propto R(\cdot)$, where $R(\cdot)$ is a black-box *reward function* or unnormalized density that takes only non-negative values. Notice that we do not know the normalizing factor $Z = \sum_{\mathbf{x}} R(\mathbf{x})$. We can use the *trajectory flow* function $F(\tau) = ZP_F(\tau)$ to take in the effect of the normalizing factor, and the corresponding *state flow* function $F(\mathbf{s}) = \sum_{\tau \ni \mathbf{s}} F(\tau)$ to model the unnormalized probability flow of intermediate state $\mathbf{s}$.

**Detailed balance (DB)**   The GFlowNet detailed balance condition provides a way to learn the above mentioned GFlowNet modules. For any single transition $(\mathbf{s} \to \mathbf{s}')$, the following DB criterion holds:

$$F(\mathbf{s})P_F(\mathbf{s}'|\mathbf{s}) = F(\mathbf{s}')P_B(\mathbf{s}|\mathbf{s}'), \quad \forall (\mathbf{s} \to \mathbf{s}') \in \mathcal{A}. \tag{3}$$

Furthermore, for any terminating state $\mathbf{x}$, we require $F(\mathbf{x}) = R(\mathbf{x})$. In practice, these constraints can be transformed into tractable training objectives, as will be shown in Section 3. Based on GFlowNet theories in Bengio et al. (2023), if the DB criterion is satisfied for any transition, then the terminating distribution $P_T(\cdot)$ will be the same desired target distribution whose density is proportional to $R(\cdot)$.

# 3 Methodology

## 3.1 Denoising Markov decision process

The denoising process for text-to-image diffusion models can be easily reformulated as a multi-step Markov decision process (MDP) with finite horizon (Fan et al., 2023; Black et al., 2023) as follows:

$$\mathbf{s}_t = (\mathbf{x}_{T-t}, \mathbf{c}), \quad p(\mathbf{s}_0) = \mathcal{N}(\mathbf{x}_T; \mathbf{0}, \mathbf{I}) \otimes p(\mathbf{c}), \quad \pi_{\boldsymbol{\theta}}(\mathbf{a}_t|\mathbf{s}_t) = p_{\boldsymbol{\theta}}(\mathbf{x}_{T-t-1}|\mathbf{x}_{T-t}, \mathbf{c}), \tag{4}$$

$$\mathbf{a}_t = \mathbf{x}_{T-t-1}, \quad r(\mathbf{s}_t, \mathbf{a}_t) = R(\mathbf{s}_{t+1}, \mathbf{c}) \text{ only if } t = T-1, \quad p(\mathbf{s}_{t+1}|\mathbf{s}_t, \mathbf{a}_t) = \delta_{\mathbf{a}_t} \otimes \delta_{\mathbf{c}}. \tag{5}$$

Here $\mathbf{s}_t, \mathbf{a}_t$ is the state and action at time step $t$ under the context of MDP. The state space is defined to be the product space (denoted by $\otimes$) of $\mathbf{x}$ in reverse time ordering and conditional prompt $\mathbf{c}$. The RL policy $\pi$ is just the denoising conditional distribution. In this MDP, when time $t$ has not reached the terminal step, we define the reward $r(\mathbf{s}_t, \mathbf{a}_t)$ to be 0. $\delta$ here denotes the Dirac distribution.

**Remark 1** (diffusion model as GFlowNet)**.** This formulation has a direct connection to the GFlowNet MDP definition in Section 2.2, which has been pointed out by Zhang et al. (2022a) and developed in Lahlou et al. (2023); Zhang et al. (2023b); Venkatraman* et al. (2024). To be specific, the action transition $(\mathbf{s}_t, \mathbf{a}_t) \to \mathbf{s}_{t+1}$ is a Dirac distribution and can be directly linked with the $(\mathbf{s}_t \to \mathbf{s}_{t+1})$ edge transition in the GFlowNet language. More importantly, the conditional distribution of the denoising process $p_{\boldsymbol{\theta}}(\mathbf{x}_{T-t-1}|\mathbf{x}_{T-t})$ corresponds to the GFlowNet forward policy $P_F(\mathbf{s}_{t+1}|\mathbf{s}_t)$, while the conditional distribution of the diffusion process $q(\mathbf{x}_{T-t}|\mathbf{x}_{T-t-1})$ corresponds to the GFlowNet backward policy $P_B(\mathbf{s}_t|\mathbf{s}_{t+1})$. Besides, $\mathbf{x}_t$ is a GFlowNet terminal state if and only if $t = 0$.

The above discussion could be summarized in the right table. In the following text, we use the denoising diffusion notation instead of GFlowNet notation as it is familiar to more broad audience. What's more, we ignore conditioning $\mathbf{c}$ for the sake of simplicity.

| Denoising diffusion | GFlowNet |
|---|---|
| $(\mathbf{x}_{T-t}, \mathbf{c})$ | $\mathbf{s}_t$ |
| $p(\mathbf{x}_{T-t-1}|\mathbf{x}_{T-t}, \mathbf{c})$ | $P_F(\mathbf{s}_{t+1}|\mathbf{s}_t)$ |
| $q(\mathbf{x}_{T-t}|\mathbf{x}_{T-t-1})$ | $P_B(\mathbf{s}_t|\mathbf{s}_{t+1})$ |

## 3.2 Diffusion alignment with GFlowNets

In this section, we describe our proposed algorithm, diffusion alignment with GFlowNets (DAG). Rather than directly optimizing the reward targets as in RL, we aim to train the generative models so that in the end they could generate objects with a probability *proportional* to the reward function: $p_{\boldsymbol{\theta}}(\mathbf{x}_0) \propto R(\mathbf{x}_0)$. To achieve this, we construct the following DB-based training objective based on Equation 3, by regressing its one side to another in the logarithm scale for any diffusion step transition $(\mathbf{x}_t, \mathbf{x}_{t-1})$.

$$\ell_{\text{DB}}(\mathbf{x}_t, \mathbf{x}_{t-1}) = (\log F_{\boldsymbol{\phi}}(\mathbf{x}_t, t) + \log p_{\boldsymbol{\theta}}(\mathbf{x}_{t-1}|\mathbf{x}_t, t) - \log F_{\boldsymbol{\phi}}(\mathbf{x}_{t-1}, t-1) - \log q(\mathbf{x}_t|\mathbf{x}_{t-1}))^2 \qquad (6)$$

We additionally force $F_{\boldsymbol{\phi}}(\mathbf{x}_t, t = 0) = R(\mathbf{x}_0)$ to introduce the reward signal. Here $\boldsymbol{\theta}, \boldsymbol{\phi}$ are the parameters of the diffusion U-Net model and the GFlowNet state flow function (which is another neural network), respectively. One can prove that if the optimization is perfect, the resulting model will generate a distribution whose density value is proportional to the reward function $R(\cdot)$ (Bengio et al., 2023; Zhang et al., 2023b).

One way to parameterize the state flow function $F$ is through the so-called forward-looking (Pan et al., 2023b, FL) technique in the way of $F_{\boldsymbol{\phi}}(\mathbf{x}_t, t) = \tilde{F}_{\boldsymbol{\phi}}(\mathbf{x}_t, t)R(\mathbf{x}_t)$, where $\tilde{F}_{\boldsymbol{\phi}}$ is the actual neural network to be learned. Intuitively, this is equivalent to initializing the state flow function to be the reward function in a functional way; therefore, learning of the state flow would become an easier task. Note that to ensure $F_{\boldsymbol{\phi}}(\mathbf{x}_0, 0) = R(\mathbf{x}_0)$, we need to force $\tilde{F}_{\boldsymbol{\phi}}(\mathbf{x}_0, 0) = 1$ for all $\mathbf{x}_0$ at the terminal step.

**Incorporating denoising diffusion-specific structure** However, the intermediate state $\mathbf{x}_t$ is noisy under our context, and thus not appropriate for being evaluated by the given reward function, which would give noisy result. What's more, what we are interested here is to "foresee" the reward of the terminal state $\mathbf{x}_0$ taken from the (partial) trajectory $\mathbf{x}_{t:0}$ starting from given $\mathbf{x}_t$. As a result, we can do the FL technique utilizing the particular structure of diffusion model as in $F_{\boldsymbol{\phi}}(\mathbf{x}_t, t) = \tilde{F}_{\boldsymbol{\phi}}(\mathbf{x}_t, t)R(\hat{\mathbf{x}}_{\boldsymbol{\theta}}(\mathbf{x}_t, t))$, where $\hat{\mathbf{x}}_{\boldsymbol{\theta}}$ is the data prediction network. We notice that a similar technique has

---

**Algorithm 1** Diffusion alignment with GFlowNets (DAG-DB & DAG-KL)

---

**Require:** Denoising policy $p_{\boldsymbol{\theta}}(\mathbf{x}_{t-1}|\mathbf{x}_t, t)$, noising policy $q(\mathbf{x}_t|\mathbf{x}_{t-1})$, flow function $F_{\boldsymbol{\phi}}(\mathbf{x}_t, t)$, black-box reward function $R(\cdot)$
1: **repeat**
2:      Rollout $\tau = \{\mathbf{x}_t\}_t$ with $p_{\boldsymbol{\theta}}(\mathbf{x}_{t-1}|\mathbf{x}_t, t)$
3:      For each transition $(\mathbf{x}_t, \mathbf{x}_{t-1}) \in \tau$:
4:      **if** algorithm is DAG-DB **then**
5:          # normal DB-based update
6:          Update $\boldsymbol{\theta}$ and $\boldsymbol{\phi}$ with Equation 8
7:      **else if** algorithm is DAG-KL **then**
8:          # KL-based update
9:          Update $\boldsymbol{\phi}$ with Equation 8
10:         Update $\boldsymbol{\theta}$ with Equation 14
11:      **end if**
12: **until** some convergence condition

---

been used to improve classifier guidance (Bansal et al., 2023). In short, our innovation in FL technique is

$$F_{\boldsymbol{\phi}}(\mathbf{x}_t, t) = \tilde{F}_{\boldsymbol{\phi}}(\mathbf{x}_t, t)R(\mathbf{x}_t) \implies F_{\boldsymbol{\phi}}(\mathbf{x}_t, t) = \tilde{F}_{\boldsymbol{\phi}}(\mathbf{x}_t, t)R(\hat{\mathbf{x}}_{\boldsymbol{\theta}}(\mathbf{x}_t, t)). \qquad (7)$$

Then the FL-DB training objective becomes

$$\ell_{\text{FL}}(\mathbf{x}_t, \mathbf{x}_{t-1}) = \left( \log \frac{\tilde{F}_{\boldsymbol{\phi}}(\mathbf{x}_t, t)R(\hat{\mathbf{x}}_{\boldsymbol{\theta}}(\mathbf{x}_t, t))p_{\boldsymbol{\theta}}(\mathbf{x}_{t-1}|\mathbf{x}_t)}{\tilde{F}_{\boldsymbol{\phi}}(\mathbf{x}_{t-1}, t-1)R(\hat{\mathbf{x}}_{\boldsymbol{\theta}}(\mathbf{x}_{t-1}, t-1))q(\mathbf{x}_t|\mathbf{x}_{t-1})} \right)^2. \qquad (8)$$

Since in this work the reward function is a black-box, the gradient flow would not go through $\hat{\mathbf{x}}_{\boldsymbol{\theta}}(\mathbf{x}_t, t)$ when we take the gradient of $\boldsymbol{\theta}$. We summarize the algorithm in Algorithm 1 and refer to it as DAG-DB.

**Remark 2** (GPU memory and the choice of GFlowNet objectives)**.** Similar to the temporal difference-$\lambda$ in RL (Sutton, 1988), it is possible to use multiple connected transition steps rather than a single transition step to construct the learning objective. Other GFlowNet objectives such as Malkin et al. (2022); Madan et al. (2022) use partial trajectories with a series of transition steps to construct the training loss and provide a different trade-off between variance and bias in credit assignment. However, for large-scale setups, this is not easy to implement, as computing policy probabilities for multiple transitions would correspondingly increase the GPU memory and computation multiple times. For example, in the Stable Diffusion setting, we could only use a batch size of 8 on each GPU for single transition computation. If we want to use a two transition based training loss, we would need to decrease the batch size by half to 4. Similarly, we will have to shorten the trajectory length by a large margin if we want to use trajectory balance. This may influence the image

generation quality and also make it tricky to compare with the RL baseline, which can be implemented with single transitions and does not need to decrease batch size or increase gradient accumulation. In practice, we find that single transition algorithms (such as our RL baseline) perform reasonably well.

**Remark 3.** In practice it makes sense to pursue a "tilt" target $p_{\boldsymbol{\theta}}(\mathbf{x}_0) \propto p_{\text{pretrained}}(\mathbf{x}_0)R(\mathbf{x}_0)$ to ensure that generated images are reasonable. This can be easily achieved within our framework by simply replacing the $q(\mathbf{x}_t|\mathbf{x}_{t-1})$ term with $p_{\text{pretrained}}(\mathbf{x}_{t-1}|\mathbf{x}_t)$ in Equation 6. We refer to Liu et al. (2024) for further details.

### 3.3 A KL-based GFlowNet algorithm with REINFORCE gradient

GFlowNet detailed balance is an off-policy algorithm that uses training data from arbitrary distributions. In this section, we derive a different KL-based on-policy objective, which has been rarely investigated in GFlowNet literature. We can reformulate DB (Equation 6) from a square loss form to a KL divergence form

$$\min_{\boldsymbol{\theta}} \mathcal{D}_{\text{KL}}\left(p_{\boldsymbol{\theta}}(\mathbf{x}_{t-1}|\mathbf{x}_t)\|\frac{F_{\boldsymbol{\phi}}(\mathbf{x}_{t-1}, t-1)q(\mathbf{x}_t|\mathbf{x}_{t-1})}{F_{\boldsymbol{\phi}}(\mathbf{x}_t, t)}\right). \tag{9}$$

In theory, when DB is perfectly satisfied, the right term $F_{\boldsymbol{\phi}}(\mathbf{x}_{t-1}, t-1)q(\mathbf{x}_t|\mathbf{x}_{t-1})/F_{\boldsymbol{\phi}}(\mathbf{x}_t, t)$ is a normalized density; in practice, it could be an unnormalized one but does not affect the optimization. Next, define

$$b(\mathbf{x}_t, \mathbf{x}_{t-1}) = \text{stop-gradient}\left(\log \frac{F_{\boldsymbol{\phi}}(\mathbf{x}_t, t)p_{\boldsymbol{\theta}}(\mathbf{x}_{t-1}|\mathbf{x}_t)}{F_{\boldsymbol{\phi}}(\mathbf{x}_{t-1}, t-1)q(\mathbf{x}_t|\mathbf{x}_{t-1})}\right), \tag{10}$$

then the KL value of Equation 9 becomes $\mathbb{E}_{p_{\boldsymbol{\theta}}(\mathbf{x}_{t-1}|\mathbf{x}_t)}[b(\mathbf{x}_t, \mathbf{x}_{t-1})]$. We have the following result for deriving a practical REINFORCE-style objective.

**Proposition 4.** *The KL term in Equation 9 has the same expected gradient with $b(\mathbf{x}_t, \mathbf{x}_{t-1})\log p_{\boldsymbol{\theta}}(\mathbf{x}_{t-1}|\mathbf{x}_t)$:*

$$\nabla_{\boldsymbol{\theta}}\mathcal{D}_{\text{KL}}\left(p_{\boldsymbol{\theta}}(\mathbf{x}_{t-1}|\mathbf{x}_t)\|\frac{F_{\boldsymbol{\phi}}(\mathbf{x}_{t-1}, t-1)q(\mathbf{x}_t|\mathbf{x}_{t-1})}{F_{\boldsymbol{\phi}}(\mathbf{x}_t, t)}\right) = \mathbb{E}_{\mathbf{x}_{t-1} \sim p_{\boldsymbol{\theta}}(\cdot|\mathbf{x}_t)}[b(\mathbf{x}_t, \mathbf{x}_{t-1})\nabla_{\boldsymbol{\theta}}\log p_{\boldsymbol{\theta}}(\mathbf{x}_{t-1}|\mathbf{x}_t)]. \tag{11}$$

We defer its proof to Section A.1 and make the following remarks:.

**Remark 5** (gradient equivalence to detailed balance)**.** Recalling Equation 6, since we have $\nabla_{\boldsymbol{\theta}}\ell_{\text{DB}}(\mathbf{x}_t, \mathbf{x}_{t-1}) = b(\mathbf{x}_t, \mathbf{x}_{t-1})\nabla_{\boldsymbol{\theta}}\log p_{\boldsymbol{\theta}}(\mathbf{x}_{t-1}|\mathbf{x}_t)$, it is clear that this KL-based objective would lead to the same expected gradient on $\boldsymbol{\theta}$ with Equation 6, if $\mathbf{x}_{t-1} \sim p_{\boldsymbol{\theta}}(\cdot|\mathbf{x}_t)$ (*i.e.*, samples being on-policy). Nonetheless, this on-policy property may not be true in practice since the current model is usually not the same as the model used for rollout trajectories after a few optimization steps.

**Remark 6** (analysis of $b(\mathbf{x}_t, \mathbf{x}_{t-1})$)**.** The term $b(\mathbf{x}_t, \mathbf{x}_{t-1})$ seems to serve as the traditional "reward" in the RL framework. Previous works (Tiapkin et al., 2024; Mohammadpour et al., 2024; Deleu et al., 2024) have shown that GFlowNet could be interpreted as solving a maximum entropy RL problem in a modified MDP, where any intermediate transition is modified to have an extra non-zero reward that equals the logarithm of GFlowNet backward policy: $r_{\text{mod}}(\mathbf{x}_t, \mathbf{x}_{t-1}) = \log q(\mathbf{x}_t|\mathbf{x}_{t-1}), t > 0$. What's more, the logarithm of the transition flow function (*i.e.*, either side of the detailed balance constraint of Equation 3) could be interpreted as the $Q$-function of this maximum entropy RL problem on the modified MDP. Therefore, we could take a more in-depth analysis of $-b(\mathbf{x}_t, \mathbf{x}_{t-1})$:

$$\underbrace{\log F_{\boldsymbol{\phi}}(\mathbf{x}_{t-1}, t-1)q(\mathbf{x}_t|\mathbf{x}_{t-1})}_{\text{modified } Q-\text{function}} + \underbrace{(-\log p_{\boldsymbol{\theta}}(\mathbf{x}_{t-1}|\mathbf{x}_t))}_{\text{entropy}} - \underbrace{\log F_{\boldsymbol{\phi}}(\mathbf{x}_t, t)}_{\text{constant}}. \tag{12}$$

The last term $F_{\boldsymbol{\phi}}(\mathbf{x}_t, t)$ is a constant for the $\mathbf{x}_{t-1} \sim p_{\boldsymbol{\theta}}(\cdot|\mathbf{x}_t)$ process. Consequently, we can see that *minimizing this KL is effectively equivalent to having a REINFORCE gradient with the modified Q-function plus an entropy regularization* (Haarnoja et al., 2018). What's more, the entropy of forward policy $\mathbb{E}_{\mathbf{x}_{t-1} \sim p_{\boldsymbol{\theta}}(\cdot|\mathbf{x}_t)}[-\log p_{\boldsymbol{\theta}}(\mathbf{x}_{t-1}|\mathbf{x}_t)]$ is actually also a constant (the diffusion model's noise schedule is fixed), which does not affect the learning.

Note that this REINFORCE style objective in Equation 11 is on-policy; the data has to come from the same distribution as the current model. In practice, the model would become not exactly on-policy after a few

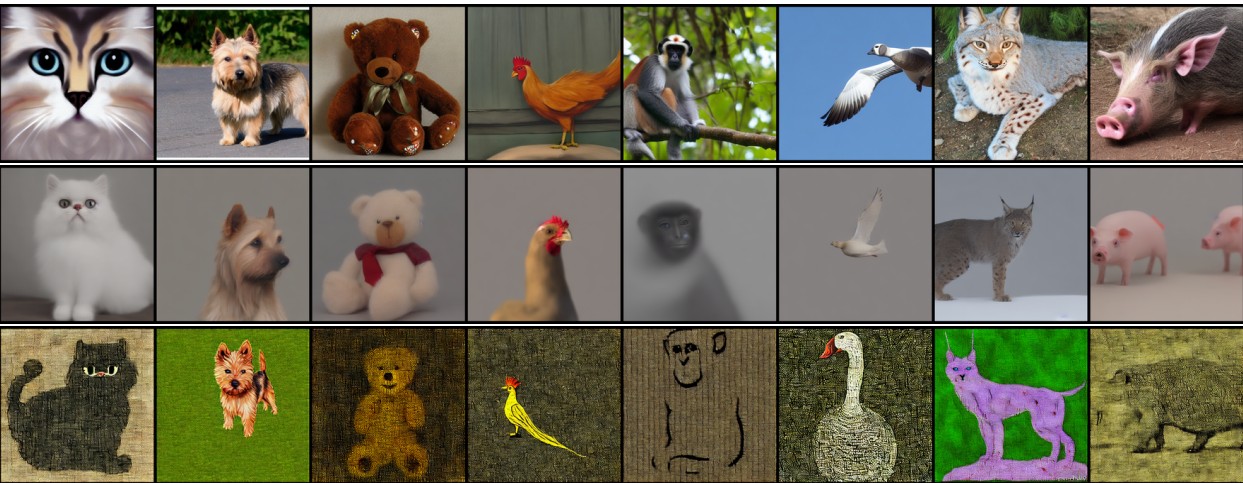

Figure 2: *Top*: samples from the original Stable Diffusion model. *Middle*: the proposed method trained with compressibility reward; these images have very smooth texture. *Down*: the proposed method trained with incompressibility reward; the texture part of images contains high frequency noise.

optimization steps, under which scenario we need to introduce the probability ratio $p_{\boldsymbol{\theta}}(\mathbf{x}_{t-1}|\mathbf{x}_t)/p_{\boldsymbol{\theta}_{\mathrm{old}}}(\mathbf{x}_{t-1}|\mathbf{x}_t)$ via importance sampling:

$$\mathbb{E}_{\mathbf{x}_{t-1} \sim p_{\boldsymbol{\theta}_{\mathrm{old}}}(\cdot|\mathbf{x}_t)} \left[ b(\mathbf{x}_t, \mathbf{x}_{t-1}) \frac{\nabla_{\boldsymbol{\theta}} p_{\boldsymbol{\theta}}(\mathbf{x}_{t-1}|\mathbf{x}_t)}{p_{\boldsymbol{\theta}_{\mathrm{old}}}(\mathbf{x}_{t-1}|\mathbf{x}_t)} \right]. \tag{13}$$

Therefore, we can define a new objective to be

$$\ell_{\mathrm{KL}}(\mathbf{x}_t, \mathbf{x}_{t-1}) = b(\mathbf{x}_t, \mathbf{x}_{t-1}) \ \mathrm{clip} \left( \frac{p_{\boldsymbol{\theta}}(\mathbf{x}_{t-1}|\mathbf{x}_t)}{p_{\boldsymbol{\theta}_{\mathrm{old}}}(\mathbf{x}_{t-1}|\mathbf{x}_t)}, 1 - \epsilon, 1 + \epsilon \right), \tag{14}$$

where $\mathbf{x}_{t-1} \sim p_{\boldsymbol{\theta}_{\mathrm{old}}}(\cdot|\mathbf{x}_t)$. Here we also introduce a clip operation to remove too drastic update, following PPO (Schulman et al., 2017). In this way, the overall gradient along a trajectory becomes

$$\mathbb{E}_{p_{\boldsymbol{\theta}_{\mathrm{old}}}(\mathbf{x}_{0:T})} \left[ \sum_{t=1}^{T} b(\mathbf{x}_t, \mathbf{x}_{t-1}) \nabla_{\boldsymbol{\theta}} \ \mathrm{clip} \left( \frac{p_{\boldsymbol{\theta}}(\mathbf{x}_{t-1}|\mathbf{x}_t)}{p_{\boldsymbol{\theta}_{\mathrm{old}}}(\mathbf{x}_{t-1}|\mathbf{x}_t)}, 1 - \epsilon, 1 + \epsilon \right) \right]. \tag{15}$$

We use this to update the policy parameter $\boldsymbol{\theta}$ and use FL-DB to only update $\phi$. We call this "diffusion alignment with GFlowNet and REINFORCE gradient" method to be DAG-KL. Note that when calculating $b(\mathbf{x}_t, \mathbf{x}_{t-1})$, we also adopt the diffusion-specific FL technique developed in Section 3.2. We also put the algorithmic pipeline of DAG-KL in Algorithm 1.

## 4 Related Works

**Diffusion alignment** People have been modeling human values to a reward function in areas such as game (Ibarz et al., 2018) and language modeling (Bai et al., 2022) to make the model more aligned. In diffusion models, early researchers used various kinds of guidance (Dhariwal & Nichol, 2021; Ho & Salimans, 2022; Kong et al., 2024) to achieve the goal of steerable generation under the reward. This approach is as simple as plug-and-play but requires querying the reward function during inference time. Another way is to post-train the model to incorporate the information from the reward function, which has a different setup from guidance methods; there is also work showing that this outperforms guidance methods (Uehara et al., 2024). Lee et al. (2023); Dong et al. (2023) achieve this through maximum likelihood estimation on model-generated samples, which are reweighted by the reward function. These works could be thought of as doing RL in one-step MDPs. Black et al. (2023); Fan et al. (2023) design RL algorithm by taking the

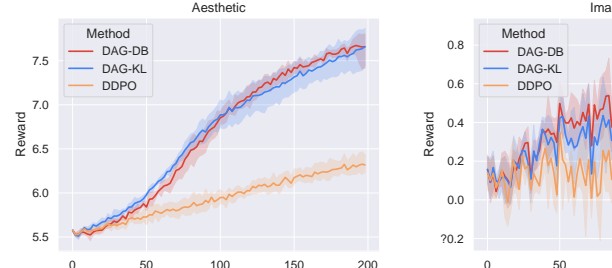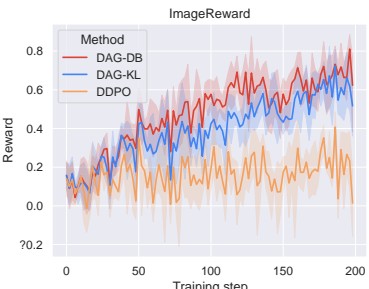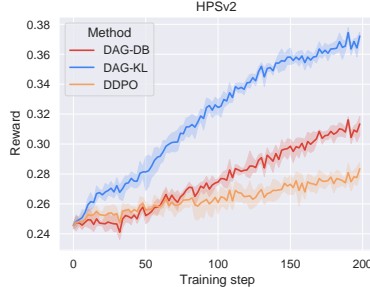

Figure 3: Sample efficiency results of our proposed methods and our RL baseline (DDPO). The number of training steps is proportional to the number of sampled trajectories. The experiments are conducted on reward functions including aesthetic score, ImageReward, and HPSv2.

diffusion generation process as a MDP (Section 3.1). In this work, we focus on black-box rewards where it is appropriate to use RL or GFlowNet methods. Furthermore, there are methods developed specifically for differentiable rewards setting (Clark et al., 2023; Wallace et al., 2023b; Prabhudesai et al., 2023; Wu et al., 2024; Xu et al., 2023; Uehara et al., 2024; Marion et al., 2024). Besides, Chen et al. (2023) study the effect of finetuning text encoder rather than diffusion U-Net. There is also work that relies on preference data rather than an explicit reward function (Wallace et al., 2023a; Yang et al., 2024). Kim et al. (2024a) investigate how to obtain a robust reward based on multiple different reward functions.

**GFlowNets** GFlowNet is a family of generalized variational inference algorithms that treats the data sampling process as a sequential decision-making one. It is useful for generating diverse and high-quality samples in structured scientific domains (Jain et al., 2022; 2023b; Liu et al., 2022; Jain et al., 2023a; Shen et al., 2023; Zhang et al., 2023e; Pan et al., 2023a; Kim et al., 2023; 2024b). A series of works have studied the connection between GFlowNets and probabilistic modeling methods (Zhang et al., 2022b; Zimmermann et al., 2022; Malkin et al., 2023; Zhang et al., 2022a; Ma et al.; Zhang et al., 2024a), and between GFlowNets and control methods (Pan et al., 2023c;d;b; Zhang et al., 2024b; Tiapkin et al., 2024). GFlowNets also have wide application in causal discovery Deleu et al. (2022), phylogenetic inference (Zhou et al., 2024), and combinatorial optimization (Zhang et al., 2023a;d). A concurrent work (Venkatraman* et al., 2024) also studies GFlowNet on diffusion alignment which is similar to this work but has different scope and different developed algorithm. Specifically, this work is aiming for posterior approximate inference that the reward function is treated as likelihood information, and develops a trajectory balance (Malkin et al., 2022) based algorithm on length modified trajectories.

## 5 Experiments

**Experimental setups** We choose Stable Diffusion v1.5 (Rombach et al., 2021) as our base generative model. For training, we use low-rank adaptation (Hu et al., 2021, LoRA) for parameter efficient computation. As for the reward functions, we do experiments with the LAION Aesthetics predictor, a neural aesthetic score trained from human feedback to give an input image an aesthetic rating. For text-image alignment rewards, we choose ImageReward (Xu et al., 2023) and human preference

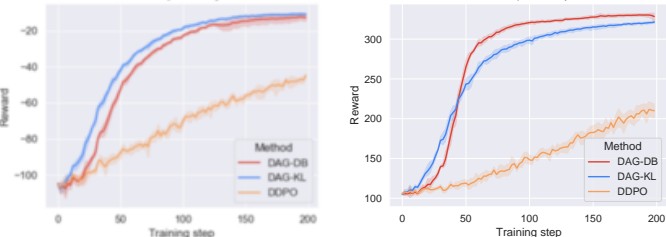

Figure 4: Sample efficiency results of our proposed methods and our RL baseline (DDPO) on learning from compressibility and incompressibility rewards.

score (HPSv2) (Wu et al., 2023). They are both CLIP (Radford et al., 2021)-type models, taking a text-image pair as input and output a scalar score about to what extent the image follows the text description. We also test with the (in)compressibility reward, which computes the file size if the input image is stored in hardware storage. As for the prompt distribution, we use a set of 45 simple animal prompts from Black et al.

A counter top with food sitting on some towels

Personal computer desk room with large glass double doors

A bathroom has a toilet and a scale

Several cars drive down the road on a cloudy day

Stable Diffusion

DDPO

DAG-DB

DAG-KL

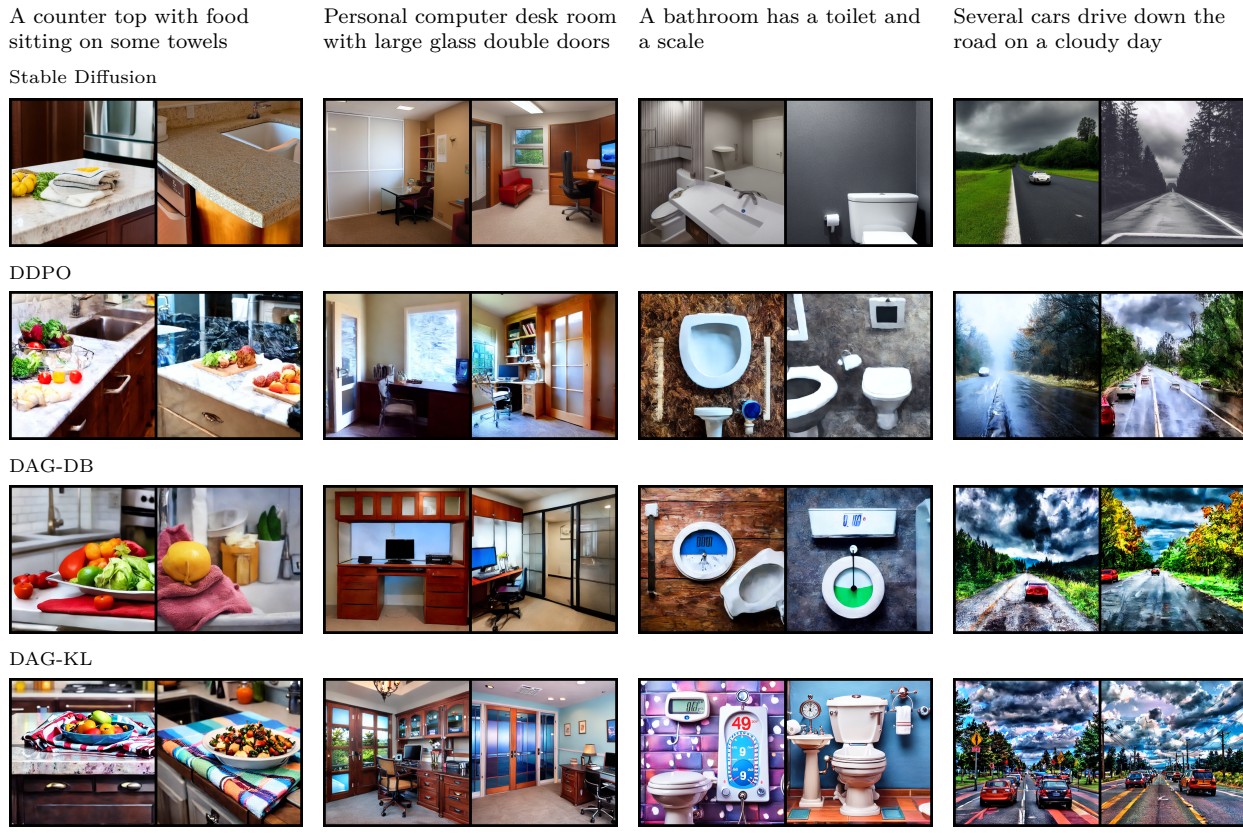

Figure 5: Text-image alignment results. We display four prompts and the corresponding generation visualization from the original Stable Diffusion (1st row), DDPO (2nd row), DAG-DB (3rd row), and DAG-KL (4th row) models to compare their alignment abilities. See Figure 10 for more results.

(2023) for the Aesthetics task; we use the whole imagenet classes for the (in)compressibility task; we use the DrawBench (Saharia et al., 2022) prompt set for the ImageReward task; we use the photo and painting prompts from the human preference dataset (HPDv2) (Wu et al., 2023) for the HPSv2 task. We notice that in our experiments, we use prompt set containing hundreds of prompts which is more than some previous work such as Black et al. (2023).

**Effectiveness of the proposed methods**    We first demonstrate that our proposed methods could generated images that have meaningful improvements corresponding to the rewards being used. In Figure 1, we compare the images from the original Stable Diffusion pretrained model and our proposed method. After our post-training, the generated images become more vibrant and vivid; we also notice that these images have slightly higher saturation, which we believe is aligned with the human preference on good-looking pictures. We also visualize the experiment results on compressibility and incompressibility tasks in Figure 2. The second row shows the generated images from the model trained with the compressibility reward, which have low details and smooth textures, and also have very limited colors. On the other hand, the model trained with incompressibility reward would generate images with high frequency texture, as shown in the third row. These results indicate that our method could effectively incorporate the reward characteristics into the generative models. We defer more experimental details to Section B.2.

**Algorithmic comparisons**    The main baseline we compare with is denoising diffusion policy optimization (Black et al., 2023, DDPO), an RL algorithm that is specifically designed for denoising diffusion alignment and has been shown to outperform other align-from-black-box-reward methods including (Lee et al., 2023; Fan et al., 2023). We show the reward curves w.r.t. the training steps of the aesthetic, ImageReward, and HPSv2 rewards in Figure 3. Here, the number of training steps corresponds proportionally to the number of trajectories collected (see appendix for more details). Both our proposed methods, DAG-DB and

| Task | Compressibility | | Incompressibility | | Aesthetic | | ImageReward | | HPSv2 | |
|---|---|---|---|---|---|---|---|---|---|---|
| Metric | Reward↑ | Div.↑ | Reward↑ | Div.↑ | Reward↑ | Div.↑ | Reward↑ | Div.↑ | Reward↑ | Div.↑ |
| DDPO | −44.74 | 27.32 | 197.83 | 25.72 | 6.33 | 13.59 | 0.29 | 26.04 | 0.29 | 19.49 |
| DAG-DB | −37.23 | 38.94 | 294.68 | 35.25 | 6.63 | 14.09 | 0.51 | 44.91 | 0.30 | 17.95 |
| DAG-KL | −34.67 | 41.30 | 218.93 | 27.68 | 6.50 | 13.63 | 0.48 | 30.37 | 0.30 | 21.85 |

Table 1: Comparison on average reward and diversity metrics across a variety of tasks. Our methods consistently achieve better trade-offs between these two directions. See Section 5 and Section B.2 for details.

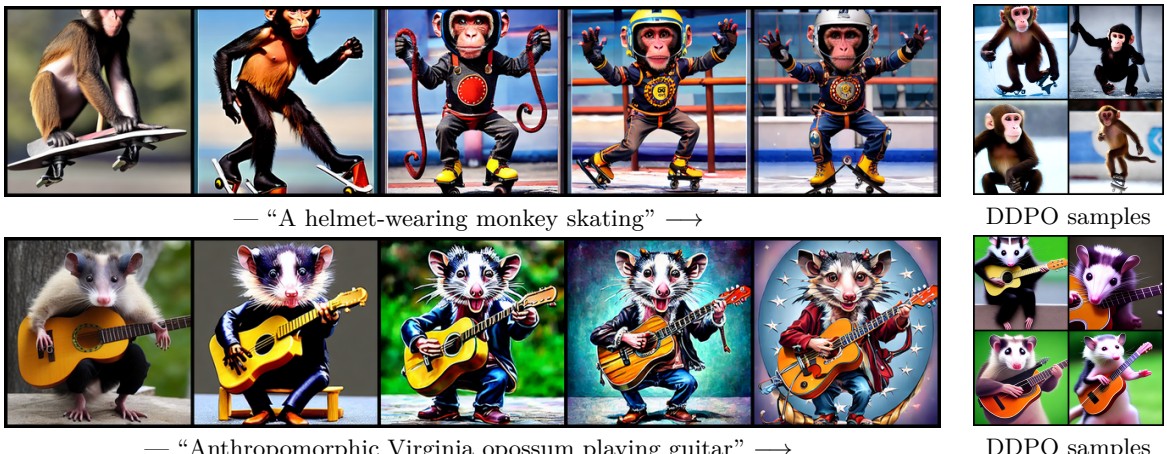

Figure 6: Visualization of alignment with regard to training progress. *Left*: the generated images from the proposed method become more aligned to the text prompt over the course of training. *Right*: samples from the DDPO baseline.

DAG-KL, achieve faster credit assignment than the DDPO baseline by a large margin. We additionally put corresponding curve plots for compressibility and incompressibility rewards in Figure 4, which also demonstrate the advantages of our proposed methods.

What's more, we provide a diversity comparison in Table 1. For the RL baseline, we take the last checkpoint; for our proposed methods, we take the earliest checkpoint with a larger reward value than the chosen RL algorithm checkpoint. This is to show that our methods could achieve results with both better reward and diversity performance. For diversity measurement, we calculate the FID score between two batches of independently generated samples from that model. We use this as a diversity metric, so the larger the better. The results indicate that GFlowNet-based methods achieve a better reward-diversity trade-off due to their distribution matching (rather than reward maximizing) formulation. We defer related details to Section B.2.

Apart from quantitative comparisons, we also visualize the alignment improvement for models trained in the HPSv2 task. In Figure 5 and Figure 10 in Appendix, we exhibit generation results for different prompts across the original Stable Diffusion, DDPO, DAG-DB, and DAG-KL models. For example, in the first "a counter top with food sitting on some towels" example, images from the original Stable Diffusion either do not have food or the food is not on towels, which is also the case for DDPO generation. This is improved for both DAG-DB and DAG-KL generation in that they capture the location relationship correctly. In the "personal computer desk room with large glass double doors" example, both the original and DDPO models cannot generate any double doors in the image, and DAG-DB model sometimes also fails. In contrast, the DAG-KL model seems to understand the concept well. Generation with other prompts also has similar results.

In Figure 6, we visualize the gradual alignment improvement of our DAG-KL method with regard to the training progress for the HPSv2 task. We show the images of our methods at 0%, 25%, 50%, 75%, and 100% training progress. In the example of "a helmet-wearing monkey skating", the DDPO baseline could generate a skating monkey but seems to fail to generate a helmet. For the proposed method, the model gradually learns

to handle the concept of a helmet over the course of training. In the "anthropomorphic Virginia opossum playing guitar" example, the baseline understands the concept of guitar well, but the generated images are not anthropomorphic, while our method manages to generate anthropomorphic opossums decently.

## 6 Conclusion

We propose Diffusion Alignment GFlowNet (DAG), a family of algorithms designed to fine-tune pretrained diffusion models based on external reward functions. Our approach advances the GFlowNet framework to accommodate the specific properties of text-to-image diffusion models. We introduce two variants, DAG-DB and DAG-KL, each tailored to optimize the model's performance with respect to different objectives. Through extensive experiments on Stable Diffusion models, we demonstrate that DAG achieves a more effective balance between maximizing reward values and maintaining output diversity than previous methods.

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

# A   Proof

## A.1   Proof of Proposition 4

*Proof.* Recalling that $b(\mathbf{x}_t, \mathbf{x}_{t-1}) = \text{stop-gradient}\left(\log \frac{F_\phi(\mathbf{x}_t, t) p_\theta(\mathbf{x}_{t-1}|\mathbf{x}_t)}{F_\phi(\mathbf{x}_{t-1}, t-1) q(\mathbf{x}_t|\mathbf{x}_{t-1})}\right)$,

$$\nabla_\theta \mathcal{D}_{\mathrm{KL}}\left(p_\theta(\mathbf{x}_{t-1}|\mathbf{x}_t) \| \frac{F_\phi(\mathbf{x}_{t-1}, t-1) q(\mathbf{x}_t|\mathbf{x}_{t-1})}{F_\phi(\mathbf{x}_t, t)}\right)$$

$$= \nabla_\theta \int p_\theta(\mathbf{x}_{t-1}|\mathbf{x}_t) \log \frac{F_\phi(\mathbf{x}_t, t) p_\theta(\mathbf{x}_{t-1}|\mathbf{x}_t)}{F_\phi(\mathbf{x}_{t-1}, t-1) q(\mathbf{x}_t|\mathbf{x}_{t-1})} \, d\mathbf{x}_{t-1}$$

$$= \int \nabla_\theta p_\theta(\mathbf{x}_{t-1}|\mathbf{x}_t) \log \frac{F_\phi(\mathbf{x}_t, t) p_\theta(\mathbf{x}_{t-1}|\mathbf{x}_t)}{F_\phi(\mathbf{x}_{t-1}, t-1) q(\mathbf{x}_t|\mathbf{x}_{t-1})} \, d\mathbf{x}_{t-1} + \int p_\theta(\mathbf{x}_{t-1}|\mathbf{x}_t) \nabla_\theta \log \frac{F_\phi(\mathbf{x}_t, t) p_\theta(\mathbf{x}_{t-1}|\mathbf{x}_t)}{F_\phi(\mathbf{x}_{t-1}, t-1) q(\mathbf{x}_t|\mathbf{x}_{t-1})} \, d\mathbf{x}_{t-1}$$

$$= \int p_\theta(\mathbf{x}_{t-1}|\mathbf{x}_t) \nabla_\theta \log p_\theta(\mathbf{x}_{t-1}|\mathbf{x}_t) b(\mathbf{x}_t, \mathbf{x}_{t-1}) \, d\mathbf{x}_{t-1} + \underbrace{\int p_\theta(\mathbf{x}_{t-1}|\mathbf{x}_t) \nabla_\theta \log p_\theta(\mathbf{x}_{t-1}|\mathbf{x}_t) \, d\mathbf{x}_{t-1}}_{=\nabla_\theta \int p_\theta(\mathbf{x}_{t-1}|\mathbf{x}_t) \, d\mathbf{x}_{t-1} = \nabla_\theta 1 = 0}$$

$$= \mathbb{E}_{\mathbf{x}_{t-1} \sim p_\theta(\cdot|\mathbf{x}_t)}\left[b(\mathbf{x}_t, \mathbf{x}_{t-1}) \nabla_\theta \log p_\theta(\mathbf{x}_{t-1}|\mathbf{x}_t)\right].$$

$\square$

# B   More about DAG

## B.1   RL optimal solutions of the denoising MDP

Training a standard RL algorithm within this diffusion MDP in Section 3.1 to perfection means the model would only generate a single trajectory with the largest reward value. This usually comes with the disadvantage of mode collapse in generated samples in practice. One direct solution is soft / maximum entropy RL (Ziebart et al., 2008; Fox et al., 2017; Haarnoja et al., 2017; Zhang et al., 2023c), whose optimal solution is a trajectory-level distribution and the probability of generating each trajectory is proportional to its trajectory cumulative reward, *i.e.*, $p_\theta(\mathbf{x}_{0:T}) \propto \sum_t R_t(\mathbf{x}_t) = R(\mathbf{x}_0)$. However, in theory this means $p_\theta(\mathbf{x}_0) = \int p_\theta(\mathbf{x}_{0:T}) \, d\mathbf{x}_{1:T} \propto \int R(\mathbf{x}_0) \, d\mathbf{x}_{1:T} = R(\mathbf{x}_0) \cdot \int 1 \, d\mathbf{x}_{1:T}$, which is not a well-defined finite term for unbounded continuous spaces. In contrast, the optimal solution of GFlowNet is $p_\theta(\mathbf{x}_0) \propto R(\mathbf{x}_0)$.

## B.2   Experimental details

Regarding training hyperparameters, we follow the DDPO github repository implementation and describe them below for completeness. We use classifier-free guidance (Ho & Salimans, 2022, CFG) with guidance weight being 5. We use a 50-step DDIM schedule. We use NVIDIA 8×A100 80GB GPUs for each task, and use a batch size of 8 per single GPU. We do 4 step gradient accumulation, which makes the essential batch size to be 256. For each "epoch", we sample 512 trajectories during the rollout phase and perform 8 optimization steps during the training phase. We train for 100 epochs. We use a $3 \times 10^{-4}$ learning rate for both the diffusion model and the flow function model without further tuning. We use the AdamW optimizer and gradient clip with the norm being 1. We set $\epsilon = 1 \times 10^{-4}$ in Equation 14. We use bfloat16 precision.

The GFlowNet framework requires the reward function to be always non-negative, so we just take the exponential of the reward to be used as the GFlowNet reward. We also set the reward exponential to $\beta = 100$ (*i.e.*, setting the distribution temperature to be 1/100). Therefore, $\log R(\cdot) = \beta R_{\mathrm{original}}(\cdot)$. Note that in GFlowNet training practice, we only need to use the logarithm of the reward rather than the original reward value. We linearly anneal $\beta$ from 0 to its maximal value in the first half of the training. We found that this almost does not change the final result but is helpful for training stability. For DAG-KL, we put the final $\beta$ coefficient on the KL gradient term. We also find using a KL regularization $\mathcal{D}_{\mathrm{KL}}(p_\theta(\mathbf{x}_{t-1}|\mathbf{x}_t) \| p_{\theta_{\mathrm{old}}}(\mathbf{x}_{t-1}|\mathbf{x}_t))$ to be helpful for stability (this is also mentioned in Fan et al. (2023)). In practice, it is essentially adding a $\ell_2$ regularization term on the output of the U-Net after CFG between the current model and previous rollout model. We simply use a coefficient 1 on this term without further tuning.

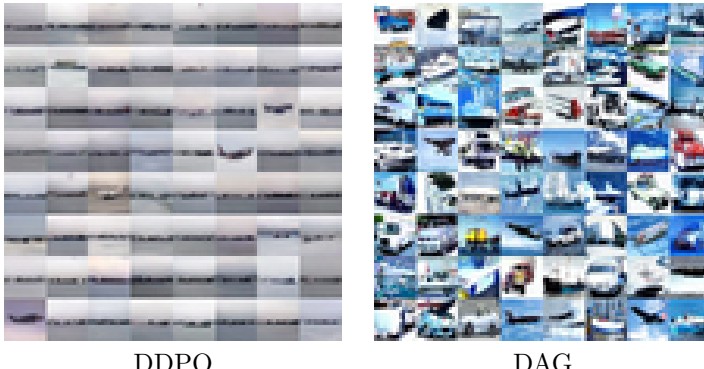

DDPM

DAG

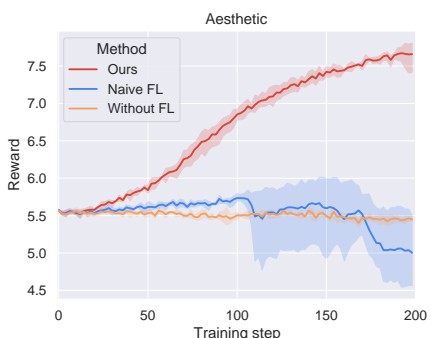

Figure 8: Samples on CIFAR-10 diffusion alignment experiments. The reward function is the probability of the generated image falling into the categories of car, truck, ship, and plane calculated by a pretrained classifier. The RL baseline shows mode collapse behaviors while the target distribution is actually multimodal.

Figure 9: Ablation study for the forward-looking (FL) usage in Section 3.2 on the Aesthetic reward task.

We use Stable Diffusion v1.5 as base model and use LoRA for post-training, following Black et al. (2023). For the architecture of the state flow function, we take a similar structure to the downsample part of the U-Net. The implementation is based on the hugging face diffusers package. We use 3 "CrossAttnDownBlock2D" blocks and 1 "DownBlock2D" and do downsampling on all of them. We set the layers per block to be 1, and set their block out channels to be $64, 128, 256, 256$. We use a final average pooling layer with kernel and stride size 4 to output a scalar given inputs including latent image, time step, and prompt embedding. We do not report diversity metric as in previous GFlowNet literature, as the average pairwise Euclidean distance in high dimensional space ($64 \times 64 \times 4 > 10,000$ dim.) is not a meaningful metric.

For computing diversity in Table 1, we take a trained model and independently generate two batches of images based on corresponding prompts, with the batch size being 2048. For our proposed methods, we choose the earliest checkpoint with a reward larger than the DDPO final rewards. We save our checkpoints for every 10 epochs. We use an Inception network to compute the mean and covariance features and calculate the Frechet distance; then we use the resulting FID as the diversity metric. Additionally, we show in Figure 7 that DAG-KL achieves a better Pareto frontier on the reward-diversity trade-off on the HPSv2 task.

### B.3 CIFAR-10 toy example

We also include a toy experiment on a CIFAR-10 pretrained DDPM[1]. We train a ResNet18 classifier and set the reward function to be the probability of the generated image falling into the categories of car, truck, ship, and plane. We use the same hyperparameters with the Stable Diffusion setting, except we only use 1 GPU with a 256 batch size for each run without gradient accumulation. We illustrate the generation results in Figure 8. We use DAG-DB here, and the DAG-KL generation is similar and non-distinguishable with it. We can see that in this relative toy task, the RL baseline easily optimizes the problem to extremes and behaves mode collapse to some extent (only generating samples of a particular plane). While for our methods, the generation results are diverse and cover different classes of vehicles. Both methods achieve average log probability larger than $-0.01$, which means the probability of falling into target categories are very close to 1.

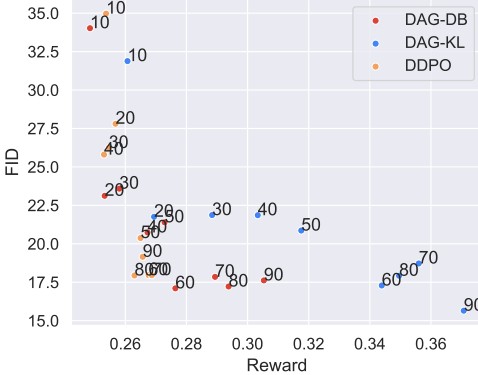

Figure 7: Scatter plot about model diversity and reward when they have been trained for different number of epochs (shown beside each of the points). It can be seen that our method (DAG-KL) achieves a better reward-diversity trade-off.

---

[1] https://huggingface.co/google/ddpm-cifar10-32

The clock on the side of the metal building is gold and black

Kitchen with a wooden kitchen island and checkered floor

A pink bicycle leaning against a fence near a river

An empty kitchen with lots of tile blue counter top space

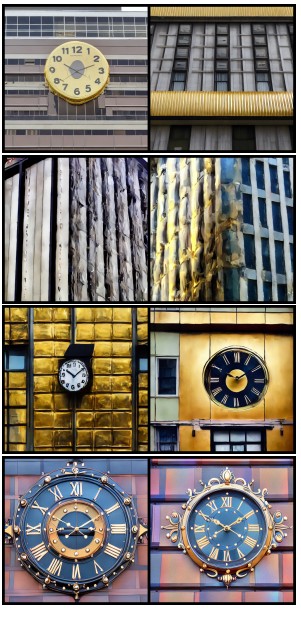
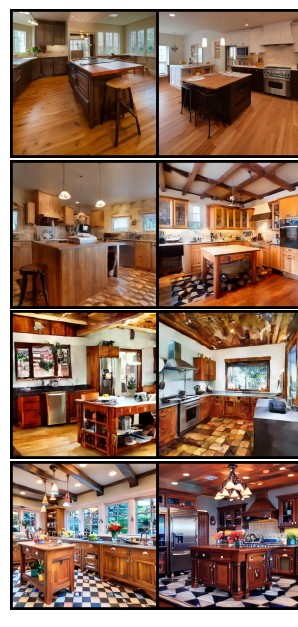
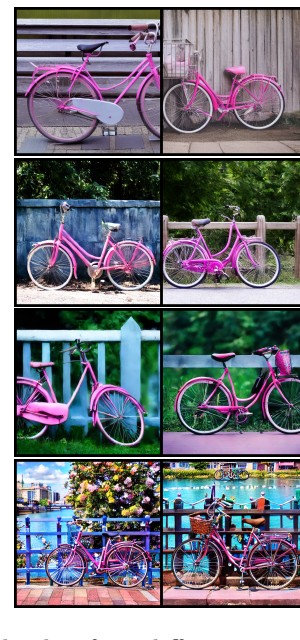
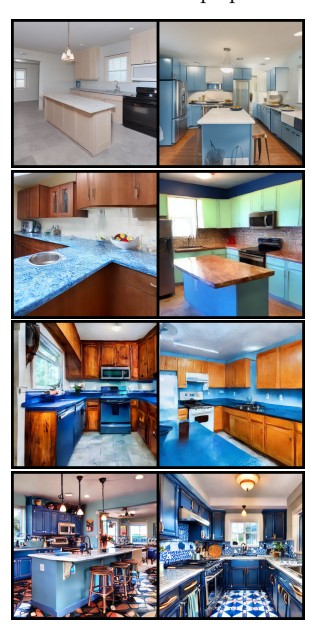

Figure 10: More text-image alignment results. We display four different prompts and the corresponding generation visualization from the original Stable Diffusion (1st row), DDPO (2nd row), DAG-DB (3rd row), and DAG-KL (4th row) models to compare their alignment ability.

## B.4 More results

In Figure 9, we perform an ablation study on the proposed denoising diffusion-specific way of forward-looking technique for the Aesthetic score task. Specifically, we compare the left and right hand sides of Equation 7, where we use "naive FL" to refer to the left hand side, and "ours" for the right hand side, as it is just the DAG-DB method in the main text. We also show the performance of not using the forward-looking technique and call it "without FL" in the figure. We can see that both the variants cannot achieve effective learning compared to our proposed method.

In Figure 10, we put more visualization comparisons about the text-image alignment performance from the models trained on the HPSv2 reward, with a similar form to Figure 5.

In Section 5 of our paper, we have discussed the reason to use the DDPO baseline as it outperforms other align-from-black-box methods including DPOK (Fan et al., 2023). For completeness, we conduct an experiment of DPOK on the Aesthetic score experiment. We keep the same experimental setup as the ones in Section 5; while DAG-DB, DAG-KL, and DDPO achives a score of 7.6587, 7.6596, and 6.3178 respectively, DPOK obtains a result of 6.1876. This result shows that our method is superior to other baselines.

