# OpenReview forum: "Improving GFlowNets for Text-to-Image Diffusion Alignment"
_TMLR — Accepted by TMLR_

### Review · Reviewer_6d2o · 2024-09-27

**Summary Of Contributions:**

This paper reformulates denoising diffusion models within the GFlowNets framework, training the model to generate samples with probabilities proportional to a reward function rather than directly maximizing it. It also reformulates the Detailed Balance (DB) equation into a KL divergence form, deriving a REINFORCE-style objective for efficient single-transition updates. This allows for larger batch sizes during training, potentially improving performance.

**Audience:**

Yes

**Claims And Evidence:**

Yes

**Requested Changes:**

How does the training efficiency and performance of the proposed method compare to recent method such as AlignProp?

Prabhudesai, Mihir, et al. "Aligning text-to-image diffusion models with reward backpropagation." arXiv preprint arXiv:2310.03739 (2023).

**Strengths And Weaknesses:**

Strengths:
1. Rigorous theoretical derivation.
2. Clear and polished presentation.
3. Strong qualitative results.
4. Reasonable quantitative performance.

Weakness:
The paper compares performance against only one baseline. Including additional comparisons with alternative alignment methods would provide a more comprehensive evaluation and further strengthen the paper’s claims.

---

> ### Author Response · Authors · 2024-11-06
>
> Thanks for the insightful review. Here is our response.
>
> > The paper compares performance against only one baseline. Including additional comparisons with alternative alignment methods would provide a more comprehensive evaluation and further strengthen the paper’s claims. How does the training efficiency and performance of the proposed method compare to recent method such as AlignProp?
>
> Thank you for the suggestion. As we have discussed in the related work section (Section 4), our paper focus on the reward finetuning problem with given **black-box** reward function. This means that we can only evaluate the value of the reward function with some input data. On the other hand, both DRaFT (Clark et al.) and AlignProp (Prabhudesai et al.) are methods that utilize the gradients of the given reward function. Therefore, in this work we do not include them in our experiments as they have a more strict assumption to the settings.
>
> To provide a more thorough experimental analysis, we add a baseline from the DPOK paper (Fan et al.) on the Aesthetic reward. We follow the same experimental setup as Section 5.
>
>
> | Methods     | DAG-DB   | DAG-KL   | DDPO     | DPOK     |
> |-------------|----------|----------|----------|----------|
> | Performance | $7.6587$ | $7.6596$ | $6.3178$ | $6.1876$ |
>
> These results show that our method is superior to other baselines under the black-box setups. We will incorporate the results into the final version of our paper.

---

### Review · Reviewer_LRTV · 2024-10-03

**Summary Of Contributions:**

This paper proposes DAG, a text-to-image diffusion model alignment method with GFlowNets rather than typical RL. DAG treats the denoising process as a MDP and improves the detailed balance (DB) objective in GFlowNets for better diffusion alignment. The authors further find the connection between DAG-DB and REINFORCE and propose DAG-KL, which updates the diffusion model and flow function independently with better alignment and diversity trade-off. The text-image samples show that the proposed DAG-DB and DAG-CL have better text-image alignments even for text with multiple concepts than DDPO.

**Audience:**

Yes

**Broader Impact Concerns:**

No broader impact concerns needed.

**Claims And Evidence:**

Yes

**Requested Changes:**

Check the listed weaknesses above. DPKO [1] (extra RL baseline) and Diffusion-DPO [2] (implicit reward baseline) could be potential choices for extra baselines.

[1] Fan, Ying, et al. "Reinforcement learning for fine-tuning text-to-image diffusion models." Advances in Neural Information Processing Systems 36 (2024).

[2] Wallace, Bram, et al. "Diffusion model alignment using direct preference optimization." Proceedings of the IEEE/CVF Conference on Computer Vision and Pattern Recognition. 2024.

**Strengths And Weaknesses:**

### Strengths
- The paper is well-organized and well-written. The motivation for using GFlowNets for diffusion alignments is well clarified, and the method is formulated clearly step-by-step.
- DAG-DB and DAG-KL show better training efficiency with better and faster convergence.
- DAG shows strong image-to-text results with improved alignment, generation diversity, and reward diversity trade-off.

### Weaknesses
- Lack of comparisons with more diffusion model alignment methods.
- A paragraph for the paper summarization is missing.
-  In Remark 2, experiments with the GFlowNets objective using longer partial trajectory lengths (e.g., SubTB with length two or more) are suggested. This would better illustrate the impact of training batch size and flow consistency over varying trajectory lengths.

---

> ### Author Response · Authors · 2024-11-06
>
> Thanks for the detailed review. We hope our response could resolve your concern. Below we post our response to each of the concerns.
>
> > Lack of comparisons with more diffusion model alignment methods.
> > DPKO [1] (extra RL baseline) and Diffusion-DPO [2] (implicit reward baseline) could be potential choices for extra baselines.
>
> Thank you for the suggestion. In page 8 of our paper, we have discussed the reason to use the DDPO baseline as it outperforms other align-from-black-box methods including DPOK. For completeness, we add the results of DPOK on the Aesthetic score experiment. We keep the same experimental setup as the ones in Section 5. These results show that our method is superior to other baselines.
>
> | Methods     | DAG-DB   | DAG-KL   | DDPO     | DPOK     |
> |-------------|----------|----------|----------|----------|
> | Performance | $7.6587$ | $7.6596$ | $6.3178$ | $6.1876$ |
>
> On the other hand, we argue that it is hard to directly compare with diffusion-DPO, as in our setup, we do not have access to pairwise preference data like in diffusion-DPO paper. What we assume is that we have an external black-box reward function that represents the preference information. We remark that we have already discussed Diffusion-DPO in the related work section in our original submission.
>
>
> > A paragraph for the paper summarization is missing.
> Thank you for the suggestion. We have added a conclusion paragraph at the end of the paper.
>
> > In Remark 2, experiments with the GFlowNets objective using longer partial trajectory lengths (e.g., SubTB with length two or more) are suggested. This would better illustrate the impact of training batch size and flow consistency over varying trajectory lengths.
>
> Thanks for the advice. We have thus compared the performance of different GFlowNet-based DAG-DB methods for the Aesthetic score. We keep the same settings as the ones in Section 5. Here we conduct experiments with sub trajectories that contain two connected transitions, opposed to the original DAG-DB that construct objectives with only one transition. We half the batch size for this variant due to the constraint of limited GPU memory.
>
> | Methods     | DAG      | DAG subtraj |
> |-------------|----------|-------------|
> | Performance | $7.6587$ | $7.6498$    |
>
> This indicates that there is no significant difference between these two variants. We believe this is due to the fact that our algorithmic innovation in Eq. 7, which incorporates the estimated reward value into the state flow function, makes the credit assignment to be less important. To put it in another way, the forward-looking trick we used in Eq. 7 shares a similar spirit to the usage of longer sub trajectories in the sense of better achieving flow consistency. Furthermore, using longer sub trajectories makes the training much slower as it needs more neural network feedforward passes, which makes it unnecessary compared to our methods based on one-transition objectives.  We will incorporate the results into the final version of our paper.

---

### Review · Reviewer_d4Ze · 2024-11-01

**Summary Of Contributions:**

In this paper, the authors propose a GFlowNet-based algorithm to post training diffusion models in a black box manner. A new objective has been proposed to improve the sample efficiency which outperformance the reinforcement learning baseline within the same number of trajectory.

**Audience:**

Yes

**Claims And Evidence:**

Yes

**Requested Changes:**

It is recommended that the authors report the quality of the generated images. Metrics like Frechet Inception Distance (FID) or other relevant image quality metrics should be included to provide a more comprehensive evaluation of the method's performance.

**Strengths And Weaknesses:**

Strengths:

* The proposed method establishes a meaningful connection between Generative Flow Networks (GFlowNets) and Diffusion Models, * which could inspire further research integrating these two frameworks for generative tasks.
* The method achieves a superior balance between reward maximization and sample diversity compared to existing state-of-the-art methods, demonstrating its effectiveness in aligning quality and diversity goals in generative modeling.

Weaknesses:

The main contributions of this work are not entirely clear. While the introduction of the KL-based on-policy objective appears to be a key innovation, the authors should further emphasize other novel aspects of the method beyond this. Clarifying additional contributions, such as specific algorithmic improvements, novel evaluation metrics, or new insights into generative modeling with GFlowNets, would strengthen the clarity and impact of the work.

---

> ### Author Response · Authors · 2024-11-06
>
> We thank the reviewer for the detailed and insightful comments. Below, we do our best to address the reviewer's questions adequately.
>
> > The main contributions of this work are not entirely clear. While the introduction of the KL-based on-policy objective appears to be a key innovation, the authors should further emphasize other novel aspects of the method beyond this. Clarifying additional contributions, such as specific algorithmic improvements, novel evaluation metrics, or new insights into generative modeling with GFlowNets, would strengthen the clarity and impact of the work.
>
> Thanks for the suggestion. We would like to restate our contributions as follows.
> First, we propose Diffusion Alignment with GFlowNet (DAG), where we utilize the special structure of denoising diffusion models to design a scalable algorithm for training diffusion models with regards to a given reward function. Second, we propose a novel KL-based objective for learning GFlowNets via importance sampling, compared to previous matching based objectives in GFlowNet literature. This not only provides empirical benefit, but also extends the original theoretical GFlowNet framework. Thirdly, the effectiveness of our proposed methodology is corroborated through large-scale experiments on Stable Diffusion.
>
> Our work provides novel insight into the intersection between the design of GFlowNet and the application of text-to-image generative models. We will make a corresponding edit into our final version.
>
> > report the quality of the generated images. Metrics like Frechet Inception Distance (FID) or other relevant image quality metrics should be included to provide a more comprehensive evaluation of the method's performance.
>
> Indeed, a good algorithm should not only increase the reward values but also maintain the visual quality. However, we would like to point out that for web-scale text-to-image generative models like Stable Diffusion, it is very hard to evaluate the image quality with a single metric such as FID. This is because FID is computed as a divergence between the distribution of generated images and a reference distribution. However, there is no directly applicable dataset for reference distribution to capture web-scale distribution for a given prompt set (see details of our choices of prompt sets in Section 5).
>
> In our work, in order to give a reasonable evaluation, we do provide both the reward value and the diversity achieved by each of the algorithms in Table 1. In Figure 7 in Appendix, we also show that our method achieves a better Pareto frontier in the reward-diversity trade-off. Notice that in Table 1 and Figure 7, we calculate the FID between two generated batches of samples to quantify the diversity of the generative models. In this way, we show that our method can both increase the achieved reward while also maintaining reasonable sample quality during the training process.

---

### Decision · Action_Editor_D5vV · 2024-12-19

**Recommendation:** Accept with minor revision

**Comment:**

This paper incorporate the diffusion model and GFlowNets frameworks, which makes post-training of diffusion models under black-box reward functions feasible. The reviewers are unanimous on the positive sides of the proposed method. Some reviewers raised some questions on the evaluation and comparing with more baselines. The authors address the issues with either more experimental results or clarifications on the concerns. The reviewers are satisfied with the rebuttal with no further concerns. I agree with the reviewers, but request the authors to incorporate additional results and further clarifications to be incorporated into the final version.

**Audience:**

Researcher in generative and diffusion models can benefit from the work.

**Claims And Evidence:**

The paper incorporates diffusion models within the GFlowNets framework, and develops a model for post-training diffusion models with a black-box reward function. It is claimed to be novel and can effectively align large-scale text-to-image diffusion models with given reward information. The algorithm development and experiment results well support the claim and are deemed sufficient by the reviewers.